# Complementarity: Toward Better Metrics and Optimizing Data Efficiency in LLMs

**Roy N. Siegelmann**  *rsiege15@jhu.edu*
*Department of Applied Mathematics and Statistics*
*Johns Hopkins University, Baltimore, MD 21218*

**Reviewed on OpenReview:** *https://openreview.net/forum?id=feAbrMXGMh*

## Abstract

Generalist Large Language Models (LLMs) are trained with an immense amount of data from across different domains. However, not all data contribute to model performance equally, and prioritizing data quality can improve domain-specific performance. We suggest that quality is not merely an independent feature of datasets, but rather the manner in which data samples interfere with or complement one another. Furthermore, existing performance metrics for language models are computationally expensive, while also frequently suffering from being mathematically ill-defined and poorly suited to generative AI. Toward improving general performance while reducing the amount of training data, and quantifying how data contributes to downstream tasks vis-a-vis their relation with other data, we introduce a new metric, Complementarity. We first establish a strong correlation between Complementarity and domain-specific task performance. Without reliance on heavy instruction-tuning and text scraping, Complementarity is significantly less expensive to compute and is applicable to a wide variety of potential target domains. Most interestingly, we demonstrate that the Complementarity taken over a training validation set provides a better predictor of generalization to future test sets than directly measuring performance on a test validation set. With this, we introduce an algorithm that carefully selects the data to fine-tune upon, leading to a high-performing fine-tuned generalist model while using only a fraction of the data, and without requiring data from the test domain. Overall, Complementarity may serve as a key metric in future analysis of data utility and design of datasets, and help facilitate the goal of a truly generalist model.

## 1 Introduction

General purpose Large Language Models (LLMs) are trained with a large corpora of datasets from a diverse set of domains. The raw amount of data is immense as in general, upsizing datasets directly impacts model performance (Brown et al., 2020; Du et al., 2021; Gururangan et al., 2020; Du et al., 2022; Kaplan et al., 2020). This presents a limitation, as finding, procuring, and training upon the amounts of data necessary to train ever-larger models (particularly those which are proficient at a wide variety of different downstream tasks) grows more expensive in time, compute, and money by the year. The energy required for training these growing models on larger-and-larger datasets is known to damage the environment and is believed to have already impacted climate change (Liu & Yin, 2024; Wu et al., 2025; Morrison et al., 2025).

The sheer amount of data being used in the training of LLMs is untenable, and experts are warning of an oncoming "data wall", wherein model training will utilize all textual data available and be left without further training resources. This is particularly an issue when it comes to multi-domain systems, which not only require a large amount of data for each domain individually (Hoffmann et al., 2022; Zeng et al., 2024), but require extra "buffer" data to guarantee that models are able to perform at a high level on all tasks, to mitigate what is sometimes referred to as catastrophic interference. Catastrophic interference is distinct from catastrophic forgetting: in traditional neural networks, catastrophic forgetting is a feature of neural networks

learning using a curriculum strategy, first one class and then another, which may cause the first class to be forgotten (McCloskey & Cohen, 1989; French, 1999). The solution to catastrophic forgetting is to combine the datasets and train together on all the data. Given that LLMs are based on likelihood maximization for text, combining datasets provides its own slate of issues, namely catastrophic interference, which is a feature of the data itself being poorly aligned and thus the model ends up learning one or all topics poorly (Li et al., 2024a; Luo et al., 2025).

Recent work suggests the importance of data quality (the token diversity, fluency, etc.) as an equally important contributor to the downstream performance of models, preventing hallucinations and enabling the models to give useful information in fluent text both for generalist (Rejeleene et al., 2024; Iyer et al., 2024; Chang et al., 2024) and domain-specific models (Bojic et al., 2023; Chen et al., 2024; Sun et al., 2024; Modesitt et al., 2024). By providing too much data, poor quality data may be mixed in; as more data is added, the average quality will eventually become lower than a subset selected for high quality. Data quality is important not only in terms of the objective attributes (reducing empty strings, random lists of symbols, profanity, typos, etc.) but in terms of relative contribution to solving tasks. While some domains complement each other and improve the overall performance of the model on target tasks, others can provide conflicting data distributions that reduce performance. Training for two deeply disparate tasks certainly may reduce the overall performance on each, but we demonstrate that data from certain domains - or certain subsets of domains - can combine with others to increase performance on both associated tasks. As such, we aim to design a metric that can be used for the selection of domains and domain subset that contribute to overall performance across different tasks, complementing each other.

We introduce a mathematically sound metric, Complementarity, which measures the impact that fine-tuning a model on source datasets from training domains has on target domain-specific task performance. We demonstrate empirically that Complementarity is strongly correlated with domain-specific task performance and thus may constitute a useful analogue for measuring shift in performance. Using Complementary, we emphasize the mutual support of different domains, which leads to a data-selection algorithm that identifies a fraction of the data to be used for training while achieving performance improvements.

Given that Complementarity does not requires generations, it is significantly faster and more computationally efficient than many state-of-the-art evaluation metrics, which require significant time and computational cost for text generation and extensive instruction prompt-tuning. Additionally, unlike many state-of-the-art metrics that require text scraping which depends on the specific model, training dataset, and target task, Complementary is task agnostic. These findings suggest Complementarity as an efficient performance metric for LLMs which can serve to complement current evaluation methods.

The paper is organized as follows. Section 2 describes the issues with state-of-the-art evaluation metrics, and presents the vision for Complementarity as a metric built to mitigate such problems. Section 3 formally introduces our Complementarity metric and its properties, and demonstrates its predictive power for downstream model performance. Section 4 constitutes our most practical contribution; it provides proof-of-concept experimental algorithm for multi-domain data selection based on Complementarity, which ends up using a fraction of the data while outperforming domain-by-domain selections and even performance-based data selection. Section 5 outlines current LLM data-selection methods. Section 6 concludes the paper by summarizing contributions and discussing avenues for future work.

## 2 Problems with Existing Metrics

There are two distinct problems with state-of-the-art metric selection in the field of LLMs, namely the cost of generations, and human time required to design text scraping and instruction-tuning for evaluation. We will describe each of them before laying out our (singular) solution to both of them in the subsequent sections.

Let us say that a researcher sets out with a new task: train a model capable of solving complicated mathematical word problems with great accuracy. To save resources and time, they decide to use an existing model and fine-tune it on a dataset that is well-aligned with this downstream task, and compare it with the baseline model. Fine-tuning goes smoothly, and the researcher is now left with two models, the base model and the math fine-tuned model. The problem now comes with how to compare the two models.

Let us zoom out a little. In the world of traditional neural networks used as classifiers, the natural manner to evaluate a new model is with basic success metrics (accuracy, precision, and F1 score come to mind among others). Feed the model ten thousand pictures of cats and dogs, and count the percentage of the time it correctly and incorrectly classified each - simple enough.

In the world of LLMs, this is somewhat more complicated. First, the generally more intractable issue, is that of evaluation speed. LLMs as generative models need to generate text to be evaluated and classified. Generating text takes time, particularly as (a) the models become larger to enhance their capabilities, with modern models easily reaching into the hundreds of billions of parameters, and (b) increasingly complicated problems require allowing for longer outputs and thus significantly more time for each generation. This is generally dealt with by simply ignoring the problem altogether and attempting to design ever more intelligent model architectures with shortcuts and industry tricks and by dedicating more computing resources to each jobs.

The second issue is one which is so deeply ingrained into the very fabric of LLMs that tens of thousands of papers have been written about it indirectly, and yet few have even addressed that it is a problem to begin with, namely the design of an evaluation metric design. In the sphere of LLMs, as language is deeply abstract, nebulous, and complicated, the notion of "Pass@1" as a single metric does not exist. Rather, the simple success metrics (accuracy, precision, pass@1, etc.) are the very tip of rather lengthy evaluation pipelines. To explain in detail, let us return to the case study of our math fine-tuned model.

Let us assume that in our mathematics dataset, we have a list of word problems along with the numeric (and **only** numeric) answer, e.g. {"Question": "The faces on a fair number die are labelled 1, 2, 3, 4, 5 and 6. You roll the die 12 times. How many times should you expect to roll a 1?", "Answer": "2"}. If you input the question into an LLM, even when asking it to answer briefly a sample response would be something like "The expected number of times to roll a 1 in 12 rolls of a fair six-sided die is: $12 \times \frac{1}{6} = \frac{12}{6} = 2$, so 2 times".

Now, while this answer is certainly correct, it is necessary to extract a number from the answer to compare with the canonical correct result. In this case, one might consider writing a RegEx text scraper to extract the last number from the answer. For this specific example this works well, but what if the answer instead reads "The expected number of times to roll a 1 in 12 rolls of a fair six-sided die is: $12 \times \frac{1}{6} = \frac{12}{6} = 2$, so **two** times", using the word "two" in place of the number? It would then be necessary to update the ReGex to also look for the last "number word" in the sentence. And what if instead of "two times" it said "twice"? The definition of "number words" would need to be expanded. This is also still all assuming that the last appearing number (whatever a number means) in the answer is the final answer, as opposed to something of the form "it is two times, since it is one-twelfth multiplied by six", which would have "six" as the final number.Another potential solution to this is instruction-tuning, i.e. giving the model specific instructions regarding how to answer a question. Telling a sufficiently powerful model to "answer with only a single number and no other text" could work. However, this relies on fine-tuning it with instructions similar to the input-output pairs, and there is still a likelihood that the LLM would not follow the exact instructions, especially if it is a smaller model.

Now, what if the exact answer is not numeric? Let us say that the question and canonical solution pair is {"Question": "Jimmy draws a geometric shape with four sides of equal length at right angles from each other. What shape did Jimmy draw?", "Answer": "Square"}. Now, while the answer is still a single word (and may even be just a single token), extracting it may be more difficult. Clearly, "square" is a nebulous concept to extract using a ReGex, so the naive solution would be to mark answers that contain "square" as correct and those which do not as incorrect. Unfortunately, this would also mark answers such as "the shape cannot be a square or a circle, so it is a triangle" as correct. The incorporation of human evaluation is potentially very expensive and highly time-consuming. Alternatively, using another model as a "judge" classifier for correctness is also time-consuming and presents additional training requirements, and which may introduce another source of bias. Thus in truth, we are most likely left with one solution: instruction-tuning - perhaps asking it to respond with only a single word - which is far from guaranteed to work.

However, another complication exists. what if the question is more of a theoretical question, such that we would require an explanation and is not a binary yes-or-no answer? Short of a human evaluator or an external classifier, the major currently-utilized way to test understanding is using multiple choice questions. This adds a whole different headache, as instruction-tuning the model to answer a multiple-choice question with only

an (a), (b), (c), or (d) requires a decently large model, a specific methodology of fine-tuning, and it would likely significantly reduce the non-multiple choice capabilities of the model. Beyond that, there is a strong chance that the model will still answer with some larger response containing one or more of (a), (b), (c), and (d), sometimes capitalized and sometimes in incorrect brackets or outside brackets overall. This means that RegEx-style text scraping is still essential, lest we bias the results by throwing out answers that are not a single answer long, potentially adding confounding factors.

In all of these cases, there is a significant amount of design time required into coming up with different instructions and text-scraping mechanisms for each different category of questions, and providing enough of each question type so that there is a sufficient breadth of data associated with each instruction as to prevent overfitting on the few examples, as instructions are learned together with the input-output pairs. Furthermore, this is is just for answering exam-style questions! What if the task is more complicated, such as asking for functional code? A solution would need to scrape away all the unnecessary answer text, compile and run the generated code, and for each example design a sufficient number of diverse test-cases to make sure it works properly.

To summarize:

1. Evaluation takes a significant amount of time due to the necessity of acquiring generations to evaluate.

2. State-of-the-Art metrics for evaluation are very difficult to apply to the LLM sphere, and require much human time.

   (a) There are various considerations for extracting the correct answer, including when and how to apply text scraping, instruction-tuning, and other tools.

   (b) Even if solutions are discovered, they are not necessarily generalizable to other questions even in the same domain, and require division, classification, manual annotation, and class balancing.

   (c) This is particularly a problem in the multi-domain case, where even if an efficient evaluation metric is discovered for one domain, other domains may differ and thus there is no replicability for general-purpose foundation models.

By introducing the metric of **Complementarity**, which does not require generations and can be used uniformly out-of-the-box across domains, we seek to ameliorate these issues. We demonstrate that Complementarity is strongly correlated with traditional evaluation metrics, and thus can serve as an alternative with the aforementioned benefits.

## 3 Complementarity Metric

This section introducess the metric of Complementarity and demonstrates that on the domains and models tested, (a) it is strongly correlated with accuracy for both models trained on single-domains and those trained on multiple domains across different downstream tasks, and (b) it is pseudo-symmetric, i.e. the Complementarity of one domain on another is largely correlated to that of the latter on the former.

### 3.1 Preliminaries and Theory

The metric of Complementarity is founded upon model perplexity. Perplexity of a model on a given piece of text is a measurement of how likely the model is to generate said text. High perplexity implies that the model assigns a low probability for the occurrence of the text, and thus is unlikely to generate it, wherein low perplexity implies that such text is likely to be generated by the model. Numerically, for text $X = (x_1, x_2, ..., x_n)$, perplexity can be denoted as follows, wherein $p(x_i \mid x_{<i})$ is the probability of a token

being generated by the model given previous tokens:

$$PP(X) = \exp\left(\frac{1}{n}\sum_{i=1}^{n} -\ln p(x_i \mid x_{<i})\right)$$

$$= \exp\left(-\frac{1}{n}\ln p\left(\prod_{i=1}^{n} x_i\right)\right)$$

$$= \frac{1}{\sqrt[n]{p\left(\prod_{i=1}^{n} x_i\right)}}$$

Building upon the established notion of perplexity, we define Complementarity as follows: For a given model $M$, and domains $D_1, ..., D_n$, denote by $M_i$ model $M$ fine-tuned on a dataset taken from domain $D_i$. Denote by $PP(D_j)$ the perplexity of domain $D_j$ when evaluated on base model $M$, and by $PP_i(D_j)$ the perplexity of domain $D_j$ when evaluated on model $M_i$. Then the Complementarity which domain $D_i$ affects upon domain (task) $D_j$ is given by

$$\mathcal{C}_{i,j} = \ln\left(\frac{PP_i(D_j)}{PP(D_j)}\right)$$

It is not trivial a priori for a change in perplexity (likelihood of generating text) to correlate with domain-specific task performance. However, there is existing theoretical backing supporting the belief that Complementarity may inform model success. First, perplexity on prompts has been recently demonstrated to correlate strongly with performance for both multiple-choice and open-ended tasks, and according to some research even more predictive of model behaviors than model size or training computation (Xia et al., 2023; Gonen et al., 2024). Second, the structure of Complementarity closely resembles an LLM version of expected information gain (Lindley, 1956), which has seen great success in decisions trees (Nowozin, 2012) and more recently in Bayesian modeling (Smith et al., 2023). Whereas expected information gain is a theoretical value that requires estimation, Complementarity is a strictly empirical correlate, and thus is efficient even when applied to generative AI (Goda et al., 2019; Li et al., 2025).

As with other perplexity-based metrics, Complementarity is sensitive to domain dependence. For example, assume we have a model trained on a coding dataset, which includes some mathematics-like expressions, such as "n = 5", "x = x + 1", "if 1 == 2". These examples familiarize the model with the given structures, and will therefore lower perplexity (i.e. raise complementarity) on a mathematics dataset. Yet, performance will likely be lowered due to the irrelevance of what was learned to mathematics. The reverse direction would not hold, as a mathematics database contains only a small percentage of example with structure relevant to coding. On the other hand, training on both mathematics and coding teaches the model to distinguish the finer features of both structures, and thus apply the correct knowledge to each problem, so Complementarity remains strong in the multi-domain case. We will see example of this in the next subsection.

## 3.2 Complementarity Predicts Success

To study the predictive capability of Complementarity as a metric, we look for a correlation between Complementarity and performance across four domains: Coding, Mathematics, Medicine, and Physics. From each domain, we select a large dataset for training (ranging from a relatively large 30,000 entries to a colossal 200,000 entries) and a well-known, widely used evaluation dataset to test knowledge in said domain. The evaluation coding dataset selected was Mostly Basic Python Problems (MBPP), which consists of an English-language prompt describing the desired behavior of a Python function, a well-annotated example function achieving the goals, and at least three test cases for evaluation. The evaluation mathematics dataset selected was Grade School Math 8k (GSM8K), which consists of a mathematics word problem, an extensive step-by-step solution, and an extracted numerical answer for evaluation. The medicine and physics datasets chosen were from Measuring Massive Multitask Language Understanding (MMLU), possibly the best-known multi-domain evaluation dataset, containing a question, four multiple choice options, and the letter of the correct answer. This collection was selected to encompass a variety of different answer types, from long

answer to numerical short response to multiple choice. For more details about the datasets used for training and evaluation are in Appendix Tables 4, 5 respectively.

For a model choice, we looked for a highly representative model, so that results will naturally generalize to many other common LLMs: one which is strong, but not over-tuned; large enough to be useful, but not so large it is difficult to deploy; one which is open-source and the base of many popular fine-tuned models; and finally is highly efficient under quantization, so it can be fine-tuned in a realistic time-scale. As such, we settled on Mistral 7B Instruct v0.2 (Jiang et al., 2023) as our main model, and LLaMa 8B Instruct (AI@Meta, 2024) for verification of our results. We created four fine-tuned versions of the base model, each on one of the four different training datasets, and evaluate the performance of each model on all four domain tasks, comparing with the Complementarity of each model on the evaluation datasets. Toward the goal of generalist models, we also studied the Complementarity metrics on models fine-tuned on datasets from multiple domains. For this, we create six fine-tuned versions of the base model, each trained on an even split from a pair of domains.

Table 1 contains the results of this experiment. It shows the strong correlation between performance and Complementarity throughout all domains and domain merges, wherein the higher the Complementarity, the better the performance, while the lower the Complementarity the worse the performance. To measure performance, we use pass@1 (P@1), i.e. the proportion of correct answers on the first try. P@1 ratio then represents the P@1 from the fine-tuned model divided by the P@1 from the base model. The high correlation of Complementarity with performance indicates we can consider Complementarity an alternative low-resource, task-agnostic performance metric. This is validated by similar results on identical experiments reported for LLaMa in Table 6 in the Appendix. We graph the correlation between Complementary and log of P@1 ratios for all ten fine-tuned models in Figure 1 to visualize their linear correlation.

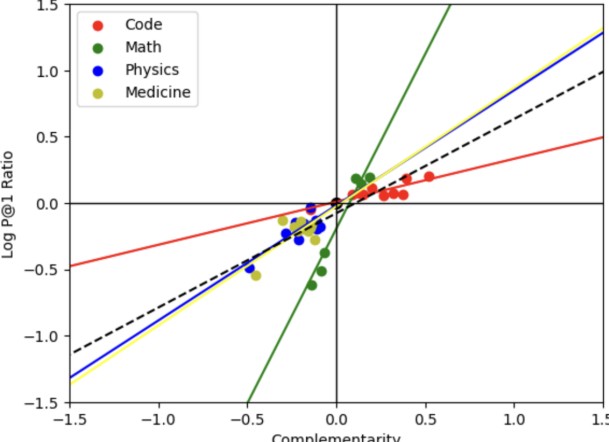

Figure 1: **Correlation between complementarity and task-specific performance.** This figure aligns with the data presented in Table 1. The correlation between log P@1 ratio and Complementarity is plotted, with the color of dots and the linear regression line representing the target evaluation domain (columns of the table). The black dashed line represents a linear regression including all points. Overall, the data relationship between Complementarity and change in accuracy appears quite linear, which is reinforced by the high linear correlation coefficients.

We note the poor performance of fine-tuned models on the physics and medicine tasks, taken from the MMLU dataset, which consists of answering multiple choice questions. This is likely a result of the fact that the training dataset did not contain any multiple choice questions. There is nevertheless a strong correlation between Complementarity and change in Pass@1, which is indicative of the strength of Complementarity. For comparison of efficiency, Pass@1 required the model to be given specific instructions for the generation formatting, which then required text-scraping to extract answers. Correctness of the text-scraping was then

Table 1: **Strong correlation between Complementarity on the test sets and task-specific performance.** A table containing the performance and Complementarity on four different tasks for models fine-tuned on the four domains and six domain merges. $\mathcal{C}$ is the Complementarity, and P@1 Ratio is the ratio of $P@1$ between the fine-tuned model and the base model. The first section is the base model followed by four rows from the model fine-tuned on a single training domain, followed by a summary of correlations between the two. The second section consists of six rows of bi-domain merged fine-tune models. The final section presents the correlation and then the p-values of all models, with statistically significant values bolded. There are two values in each of the correlation computations for the math task, wherein the second value is derived from removing the code and code-based combination models from the tally due to conditional domain dependence between the coding training dataset and mathematics evaluation dataset. Correlation between performance and Complementarity is statistically significant for all tasks. When taking the entire table into account, there is an average correlation of 0.7603 (0.8599 when discounting the code models on the math task), which represents a strong p-value. This constitutes support for Complementarity being a predictor of performance.

| Target / Source | Code (MBPP) | | Math (GSM8K) | | Medicine (MMLU) | | Physics (MMLU) | |
|---|---|---|---|---|---|---|---|---|
| | $\mathcal{C}$ | P@1 Ratio | $\mathcal{C}$ | P@1 Ratio | $\mathcal{C}$ | P@1 Ratio | $\mathcal{C}$ | P@1 Ratio |
| Base | 0 | 1 | 0 | 1 | 0 | 1 | 0 | 1 |
| Code | 0.40 | 1.21 | 0.06 | 0.24 | -0.30 | 0.88 | -0.28 | 0.79 |
| Math | 0.27 | 1.06 | 0.19 | 1.21 | -0.22 | 0.81 | -0.20 | 0.86 |
| Medicine | -0.14 | 0.95 | -0.14 | 0.54 | -0.45 | 0.58 | -0.49 | 0.62 |
| Physics | 0.09 | 1.06 | -0.07 | 0.69 | -0.20 | 0.87 | -0.15 | 0.97 |
| Correlation (Single) | — | 0.94 | — | 0.27 / 0.93 | — | 0.88 | — | 0.97 |
| Code+Math | 0.52 | 1.22 | 0.19 | 1.31 | -0.14 | 0.85 | -0.11 | 0.82 |
| Code+Medicine | 0.38 | 1.06 | 0.02 | 0.39 | -0.12 | 0.76 | -0.18 | 0.86 |
| Code+Physics | 0.32 | 1.08 | -0.04 | 0.45 | -0.24 | 0.84 | -0.21 | 0.76 |
| Math+Medicine | 0.13 | 1.08 | 0.14 | 1.15 | -0.19 | 0.82 | -0.23 | 0.86 |
| Math+Physics | 0.20 | 1.12 | 0.11 | 1.20 | -0.14 | 0.83 | -0.09 | 0.83 |
| Medicine+Physics | 0.15 | 1.07 | -0.08 | 0.60 | -0.15 | 0.81 | -0.11 | 0.87 |
| Correlation | — | 0.86 | — | 0.55 / 0.95 | — | 0.77 | — | 0.86 |
| p-value | — | **1.5e-3** | — | 1e-1 / **3.1e-5** | — | **8.8e-3** | — | **1.3e-3** |

determined to be consistent by comparison with human evaluation. On the other hand, Complementarity did not require either process.

Complementarity, as seen in Table 2, is pseudo-symmetric metric, i.e. the Complementarity of one domain on another is closely related to the Complementarity of the latter domain on the first. The diagonal is positive, since fine-tuning on a dataset from a given domain increases its fluency on the test set of the same domain. The Complementarity of code on medicine is similar to that of medicine on code, and the same occurs for all other domain pairs. Taking the Pearson correlation of cross-diagonal entries yields $r = 0.838$, a strong correlation which indicates that Complementarity is indeed pseudo-symmetric. This is again validated on LLaMa in Table 7 in the Appendix. It is relatively unsurprising that for single domains, medicine has a negative complementarity on all other domains, and the other domains have a negative complementarity on medicine, given that the other three domains make greater use of quantitative answers and mathematical calculation. In particular, code and medicine impact each other poorly as single domains, given that the structure of medical and code texts are quite different. The Complementarity between each target domain and the models trained on each training domain and domain pair are visualized in Figure 2.

Pseudo-symmetry is a useful analytic property, and indicates that Complementarity represents a fundamental attribute of the underlying data, rather than the specifics of the model. Linguistic domains, particularly in high-dimensional representations, are known to generally cluster into defined regions of the representation space (Aharoni & Goldberg, 2020; Gururangan et al., 2023). Although Complementarity is not fully symmetric due to differences across domain pairs regarding utilization of tokens, prevalence of n-grams, etc., the process

of learning to align language with the target structure and gain fluency in a particular section of the language space is a bi-directional process, such that training on either domain will similarly impact Complementarity.

We observe in Table 2, that domain merges including the domain itself are generally better than the single domain for all domains but mathematics. This likely holds due to providing contrast, such that the model better learns to distinguish problems from each particular domain. In fact, other than evaluation on the mathematics dataset, all domain merges on all domain evaluations perform better than either component part, e.g. code+physics performing better on medicine than either code or physics. Regarding mathematics, perplexity is not a natural measure, reflected in the historically poor computational skills of LLMs (Xu & Ma, 2025; Boye & Moell, 2025; Bertolazzi et al., 2025), given that equations do not follow the general structure of natural language. This, we hypothesize, is why domain merges with mathematics do not contribute to complementarity beyond mathematics alone.

Table 2: **Pseudo-Symmetry of complementarity and utility of data mixing.** The first section of the table contains the Complementarity on test splits of the four training datasets for models fine-tuned on the four domains and six domain merges, and colored based on the strength of Complementarity. We can observe that Complementarity is a pseudo-symmetric metric, with a correlation coefficient between corresponding elements across the diagonal of r = 0.838, resulting in a p-value of 0.03724, a statistically significant correlation. The second section presents the Complementarity on the same test sets for models fine-tuned on training domain pairs. The highest Complementarity for each evaluation domain (column) is bolded, and the second-highest is underlined. This suggests that mixing datasets achieves similar (if not better) performance for the selected domains as those trained solely on one of them without sacrificing performance on other domains. For example, the model fine-tuned on a mix of medicine and physics data results in higher Complementarity on the medicine domain than the model fine-tuned solely on medicine, and higher Complementarity on the physics domain than the model fine-tuned solely on physics. This finding hints that a reduction of data quantity may not harm performance, and that generalist models can be sufficiently competitive with specialist models, reducing the quantity of required models.

| Source \ Target | Code | Math | Med | Phys |
|---|---|---|---|---|
| Code | 0.167 | -0.024 | -0.262 | 0.036 |
| Math | -0.072 | **0.285** | -0.106 | -0.011 |
| Medicine | -0.221 | -0.130 | 0.100 | -0.048 |
| Physics | -0.124 | -0.057 | -0.106 | 0.033 |
| Code+Math | **0.202** | 0.245 | -0.035 | 0.053 |
| Code+Medicine | 0.193 | -0.051 | **0.266** | 0.037 |
| Code+Physics | 0.159 | -0.070 | -0.141 | 0.052 |
| Math+Medicine | -0.067 | 0.227 | 0.244 | -0.006 |
| Math+Physics | 0.001 | 0.226 | -0.023 | 0.053 |
| Medicine+Physics | -0.020 | -0.053 | 0.234 | **0.073** |

## 4 Data Selection for Model Training with Complementarity

Now that the correlation between Complementarity and performance has been solidified on a broad selection of datasets, we turn to using Complementarity as a metric for training data selection. While Complementarity on a test set is correlated with performance on the same set, we may not have access to a robust test set for a given domain, nor would we want to overfit the model by optimizing for performance on the test set even if we did. Instead, we utilize Complementarity only on parts of our training corpus designated for validation to optimize model performance. Our proof-of-concept algorithm is as follows:

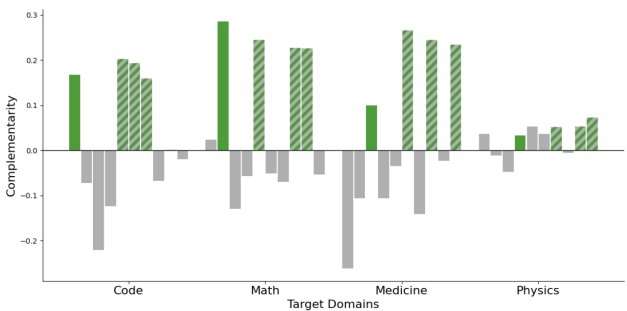

Figure 2: **Complementarity on training domains.** This figure aligns with the data presented in Table 2. The Complementarity of each fine-tuned model on different training domains is plotted in the same order as in the table. Green bars represent that the source and target datasets align, the grey bars represent no overlap, and the green-grey dashed bars represent that the model was trained on a pair of datasets including the target.

> Given a base model $M$ and $p$ domains, split off training data $R_1, ..., R_p$ and validation sets $V_1, ... V_p$.
>
> 1. Create $N$ splits $\tilde{R}_1, ..., \tilde{R}_N$ of combined $R_1, ..., R_p$ dataset. Fine-tune to receive $\tilde{M}'_i s$.
>
> 2. Select the $k$ $\tilde{M}'_i s$ with highest averaged Complementarity across all $V_1, ..., V_p$.
>
> 3. Combine the associated split datasets and fine-tune $M$, receiving $M^*_{avg}$ as the choice model.

One can observe that the amount of training required is roughly comparable to the full model, albeit more parallelizable. We next compare the algorithm's results with other possible comparison metrics which are defined as follows.

- **Calculating** $M^*_i$: For each domain $D_i$, select the $k$ fine-tuned models with the highest Complementarity on $V_i$. Combine $k$ associated split datasets and fine-tune $M$ on the combined dataset.

- **Calculating** $M^{test}_i$: For each domain, take a test set $T_i$ with performance metrics. Repeat the above but choose the highest performing fine-tuned models on test performance.

- **Calculating** $M^{test}_{avg}$: Calculate analogously to $M^*_{avg}$ but select the fine-tuned models with the highest average testing performance.

Table 3 summarizes the results for $p = 4$, $k = 2$, and $N = 10$, using the same four domains as above. It compares performance, with P@1 being pass-at-1 on domain-specific tasks aligning with the mentioned domain and dataset, and Ratio being the relative change in performance compared to the base model. The best fine-tuned performance in each column is bolded. The table supports three findings: (a) In the realm of general-purpose LLMs small datasets may be just as useful, or even more useful than large datasets, depending on the Complementarity of the data. (b) A model trained to perform well at multiple different tasks (domains) may perform even better than models trained to be experts at said domain. (c) Complementarity, on a split from the training data, may prove more useful in predicting test performance than directly measuring performance on test data. This last finding supports an efficient data-selection algorithm which uses Complementarity as its decision metric on data solely split from the training set.

Table 3: **Using Complementarity on validation for training data selection yields model improvement.** A table presenting the results of the experiment for $p = 4$, $k = 2$, and $N = 10$. Performance of different models on each of the four tasks is depicted, with P@1 being pass-at-1 on domain-specific tasks aligning with the mentioned domain and dataset, and RC being the relative change in performance compared to the base model. The "Average" column represents the average RC across the four domains. The best fine-tuned model performance in each column is bolded. For comparison, the first section consists of the baseline model and the model fine-tuned naively on all the data, i.e. 10,000 entries from each of the four domains. The second section consists of models fine-tuned on data selected by evaluating Complementarity. The first row are four different models fine-tuned on the best two 1/10ths of the dataset in terms of in-domain Complementarity. For example, for the physics evaluation, we first divide the combined dataset into ten subsets and fine-tune a model on each. Then, we compute the Complementarity scores on a validation split from the physics training data, and select the two highest Complementarity splits to use for training the model for physics. This is repeated across the four domains, and thus this row has a different model for each domain. The next row corresponds to averaging the model fine-tuned on the two splits out of ten with the best Complementarity averaged across all domains. The final section consists of models fine-tuned on data selected by evaluating performance on the test set, parallel to the previous section. We see that the model fine-tuned on data splits selected by average Complementarity on the validation set not only outperforms the model fine-tuned on all data, but outperforms or ties the models trained based on domain-specific Complementarity, and all models trained on test set performance.

| Target
Source | Code (MBPP) | | Math (GSM8K) | | Medicine (MMLU) | | Physics (MMLU) | | Average |
|---|---|---|---|---|---|---|---|---|---|
| | P@1 | Ratio | P@1 | Ratio | P@1 | Ratio | P@1 | Ratio | |
| $M$ on all $T_i$ | 24.23 | 1 | 40.33 | 1 | 65.57 | 1 | 45.53 | 1 | 1 |
| $M_{all}$ on all $T_i$ | 25.87 | 1.07 | 46.78 | 1.16 | 52.35 | 0.80 | 38.20 | 0.84 | 0.965 |
| $M_i^*$ on paired $T_i$ | **26.80** | **1.11** | 46.85 | 1.15 | 58.28 | 0.89 | 38.46 | 0.84 | 0.998 |
| $M_{avg}^*$ on all $T_i$ | **26.80** | **1.11** | **46.93** | **1.16** | **59.18** | **0.90** | **42.95** | **0.94** | **1.028** |
| $M_i^{test}$ on paired $T_i$ | 25.26 | 1.04 | 41.85 | 1.04 | 56.67 | 0.86 | **42.95** | **0.94** | 0.970 |
| $M_{avg}^{test}$ on all $T_i$ | 25.67 | 1.06 | 44.05 | 1.09 | 55.51 | 0.85 | **42.95** | **0.94** | 0.985 |

## 5 Related Work

Research in the topic of data selection is of immense quantity and continues to be of great interest (Albalak et al., 2024; Li et al., 2024b; Zhang et al., 2025b;a). Arguably the most common path for raising data quality is through data heuristics - filtering out objectively bad data - e.g. elements with less than a certain number of characters, words, or tokens (Raffel et al., 2023). Repetition count is also used, removing elements that repeat certain tokens or words too frequently in quick succession (Raffel et al., 2023), or just removing all data elements which end with, begin with, or include particular undesirable tokens (Penedo et al., 2023). Perhaps the most frequent of such methodologies depend on statistics, such as removing elements with a word count too many standard deviations above the corpus mean, those with too high a percentage of uppercase letters or symbols, or those which have a mean or standard deviation above a pre-established hard limit (Rae et al., 2022; Chen et al., 2021).

Another direction, more closely related to ours, are those of selecting data which either correlate most strongly with data from a given domain or which are thought to improve performance on an eventual downstream task (Axelrod, 2017; Feng et al., 2022; Xie et al., 2023b; Engstrom et al., 2024). But these efforts carry the unfortunate consequence of optimizing for one task or domain at the expense of others, causing catastrophic interference, reducing overall performance on other downstream tasks.

The natural solution to this is data mixing - taking data from different domains (or targeted at different downstream tasks) to form one large dataset upon which to train. As one might expect, relative percentages of data drawn from each dataset are shown to correlate strongly with downstream performance on different domain-related tasks (Xie et al., 2023a; Albalak et al., 2023; Fan et al., 2024; Xia et al., 2024). Many such methods are offline, but there are also training-time variants that continuously update inter-domain

information as training weights for how much data to draw from different domains (Chen et al., 2023). This is, however, without the more fine-grained data selection from within domains as we do.

Nevertheless, all aforemention methods rely on inputting immense amounts of data from all relevant domains, including additional buffer data to separate domain pairs to reduce catastrophic interference. One work (Azeemi et al., 2023) suggests that pruning the data can actually result in improved results with a significant reduction in overall quantity, as we also found to be the case. We refine this angle, introducing a powerful metric which can be used to guide data pruning for increased performance along a wide variety of domains and downstream tasks while significantly reducing the overall amount of data required by the model.

## 6 Discussion

Selection of a metric is an incredibly important task when it comes to evaluation of a new model or novel methodology. In many ways, the metric selected is a reflection of the user's viewpoint, informing how they will observe information. To reduce biases, selected metrics should be well grounded, i.e. have an intrinsic connection to the real world and be directly informed by the desired model behavior. This entails, in particular, selecting a metric that is mathematically similar to a fundamental property of the system, e.g. the average of five consecutive samples is relevant in an neural network with a window of five items but not in a network with a window of seven.

Many performance metrics used in the LLM sphere are inherited from classifier neural networks, and are thus ill-suited to evaluate generative AI. Indeed, while basic accuracy is mathematically simple, sound, and well-grounded metric for classifiers, what we call 'accuracy' in LLMs is achieved by a complex combination of accuracy with text scraping, instruction-tuning, and a variety of other methods which allow answer extraction. This overall accuracy metric is deeply process-dependent and thus lacks a meaningful and intuitive connection to the real world. Beyond this, accuracy and similar process-dependent metrics are computationally expensive and require case-by-case design and may depend on the model, training dataset, evaluation dataset, downstream task, etc. We introduce Complementarity, a sound metric that is calculated directly in a generation-free manner and which is demonstrated to correlate with Pass@1 on sample domains with distinct tasks.

Using Complementarity in place of domain-specific metrics for measuring performance presents three key benefits. First and foremost, Complementarity is a more robust measure than many state-of-the-art metrics. Traditional metrics such as Pass@N, accuracy, and precision operate as binary correctness measures, judging solely whether a final solution is correct or incorrect, whereas Complementarity functions as a continuous fluency measure, evaluating the extent to which outputs align with the target domain. Similarly to how students will receive partial credit for the structure and design process, so too Complementarity presents a metric for measuring understanding which is robust to typos and arithmetic errors. Second, Complementarity (which boils down to computing loss) is much faster and less compute-heavy than generation, not to mention the process used to evaluate the generations. Finally, on the topic of generation evaluation, Complementarity reduces the need for extensive prompt tuning and design of generation-scraping tools. Whereas to evaluate performance on a dataset of multiple-choice question, one of math word problems, or one of coding prompts may require prompt tuning to receive generations in a very specific format and then designing a text scraper to extract the answers (which may not even be correctly formatted), Complementarity requires no such design, being applicable to a wide variety of datasets and more grounded.

Conventional wisdom about general-purpose LLMs is that they are simply an extension of specialist models. If a model trained to perform mathematical calculations requires one million examples, a model trained to perform five tasks will simply require one million examples per task as though we were training five separate specialist models in one, with some extra data "buffer" to keep the tasks nearly independent. This, as we demonstrate, is not necessary. In training our models, we achieve better performance by fine-tuning on a well-selected 1/5th of the data than is achieved by the model fine-tuned on all the data. Thus, data quantity is not the limiting factor, and there must be some element of data quality which informs the performance of a generalist model on different tasks. We also find in our experiments that models trained on data selected by Complementarity outperformed those trained on data selected by traditional evaluation metrics, exhibiting the strength of the metric in the important task of data selection. We further show that generalist models

trained on optimizing Complementarity across the different studied domains outperform even the models specifically trained for those tasks, indicating that the data selected truly do 'complement' each other.

There are a few important angles we believe should be pursued in future research. First and foremost is the fact that Complementarity, by taking the difference in entropy from a base model to the fine-tuned model, is inherently conditioned upon the data used for pretraining the base model. Thus, we hypothesize that Complementarity would be even more useful as a metric for selecting pre-training data than it has been for fine-tuning. Another direction is that of metric merging. Ultimately, both fluency and correctness are important, and thus using a choice selection of accuracy-based traditional metrics to aid the fluency-based Complementarity in data selection may provide a benefit over using either in isolation. Lastly, the burden of answer extraction and numerical assessment makes certain datasets and domains infeasible for evaluation, such as creative writing and poetry which have no singular correct response. Since Complementarity is not a binary metric and rather generally measures fluency, we hypothesize that this difficulty may be somewhat mitigated by the use of Complementarity.

We suggest that Complementarity is a tool which may be able to enhance model capabilities while decreasing cost and time in data harvesting, design, and compute. This offers the potential of facilitating the advent of generalist foundation models. Lastly, we believe Complementarity may yet provide the basis for a metric paradigm shift, wherein the extensive design of convoluted metrics stemming from classifiers is phased out in favor of intuitive, efficient, and mathematically sound metrics.

## Acknowledgements

We thank Ashwin Ganesan and Hava Siegelmann for their contributions to this manuscript, between brainstorming ideas, expanding the scope of hypothesis, and providing help in editing this text. We further thank Prithviraj Tarale for help in style adherence.

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

## Appendix

Table 4: **Training datasets.** The four datasets used for training, aligning with the four domains of coding, mathematics, medicine, and physics respectively, are named and basic statistics about the data is provided.

| Dataset | Samples | # Words per Prompt | | # Words per Response | |
|---|---|---|---|---|---|
| | | Mean | STD | Mean | STD |
| Code Instructions 120k | 121,959 | 92.11 | 73.62 | 425.97 | 691.09 |
| ORCA Math Word Problems 200k | 200,035 | 238.87 | 145.03 | 878.43 | 408.94 |
| Medical Meadow Medical Flashcards | 33,955 | 92.4 | 36.27 | 349.10 | 313.55 |
| Physics Instruct Tune 30k | 30,231 | 43.81 | 17.02 | 646.22 | 577.82 |

Table 5: **Test datasets.** The four datasets used for evaluation, aligning with the four domains of coding, mathematics, medicine, and physics respectively, are named and basic statistics about the data is provided.

| Dataset | Samples | # Words per Prompt | | # Words per Response | |
|---|---|---|---|---|---|
| | | Mean | STD | Mean | STD |
| Mostly Basic Python Problems (MBPP) | 974 | 78.62 | 21.62 | 181.07 | 127.42 |
| Grade School Math 8K (GSM8K) | 1319 | 239.87 | 97.57 | 292.88 | 141.77 |
| MMLU: Medicine | 481 | 186.03 | 463.49 | 13.75 | 3.9 |
| MMLU: Physics | 557 | 142.1 | 91.33 | 56.5 | 0.87 |

Table 6: **Strong correlation between Complementarity on the test sets and task-specific performance: LLaMa.** A table containing the performance and Complementarity on four different tasks for models fine-tuned on the four domains and six domain merges. The contents are similar to Table 1, but run on LLaMa instead of Mistral. Once again, the correlation between performance and Complementarity is statistically significant for all tasks. When taking the entire table into account, there is an average correlation of 0.7302 (0.8277 when discounting the code models on the math task), which represents a strong p-value.

| Target / Source | Code (MBPP) | | Math (GSM8K) | | Medicine (MMLU) | | Physics (MMLU) | |
|---|---|---|---|---|---|---|---|---|
| | $\mathcal{C}$ | P@1 Ratio | $\mathcal{C}$ | P@1 Ratio | $\mathcal{C}$ | P@1 Ratio | $\mathcal{C}$ | P@1 Ratio |
| Base | 0 | 1 | 0 | 1 | 0 | 1 | 0 | 1 |
| Code | 0.58 | 1.24 | 0.12 | 0.18 | -0.14 | 0.81 | -0.18 | 0.77 |
| Math | 0.08 | 1.07 | 0.25 | 1.17 | -0.20 | 0.77 | -0.13 | 0.89 |
| Medicine | -0.22 | 0.92 | -0.17 | 0.57 | -0.21 | 0.49 | -0.20 | 0.64 |
| Physics | 0.12 | 1.02 | -0.06 | 0.62 | -0.20 | 0.79 | -0.11 | 0.92 |
| Correlation (Single) | — | 0.98 | — | 0.32 / 0.89 | — | 0.79 | — | 0.91 |
| Code+Math | 0.53 | 1.28 | 0.20 | 1.40 | -0.13 | 0.82 | -0.22 | 0.77 |
| Code+Medicine | 0.19 | 1.03 | 0.04 | 0.34 | -0.25 | 0.71 | -0.09 | 0.84 |
| Code+Physics | 0.56 | 1.11 | 0.09 | 0.44 | -0.10 | 0.82 | -0.16 | 0.73 |
| Math+Medicine | 0.38 | 1.08 | 0.12 | 1.20 | -0.13 | 0.76 | -0.12 | 0.80 |
| Math+Physics | 0.25 | 1.14 | 0.14 | 1.16 | -0.17 | 0.74 | -0.10 | 0.85 |
| Medicine+Physics | 0.08 | 1.06 | -0.08 | 0.55 | -0.15 | 0.84 | -0.17 | 0.86 |
| Correlation | — | 0.85 | — | 0.51 / 0.90 | — | 0.74 | — | 0.82 |
| p-value | — | **1.8e-3** | — | 1.3e-1 / **3.8e-5** | — | **9.8e-3** | — | **3.6e-3** |

Table 7: **Pseudo-Symmetry of complementarity and utility of data mixing: LLaMa.** This table contains the Complementarity on test splits of the four training datasets for models fine-tuned on the four domains followed by the six domain merges. The contents are similar to Table 2, while the run is on LLaMa instead of Mistral. We again can observe that Complementarity is a pseudo-symmetric metric, with a correlation coefficient between corresponding elements across the diagonal of r = 0.844, resulting in a p-value of 0.0170, a strong and statistically significant correlation.

| Target / Source | Code | Math | Med | Phys |
|---|---|---|---|---|
| Code | 0.148 | -0.052 | -0.280 | 0.031 |
| Math | -0.034 | **0.253** | -0.101 | 0.088 |
| Medicine | -0.257 | -0.210 | 0.113 | -0.066 |
| Physics | -0.117 | 0.018 | -0.072 | 0.041 |
| Code+Math | **0.189** | 0.244 | -0.040 | 0.048 |
| Code+Medicine | 0.151 | -0.046 | 0.248 | 0.022 |
| Code+Physics | 0.162 | -0.037 | -0.155 | 0.057 |
| Math+Medicine | -0.022 | 0.208 | 0.211 | 0.005 |
| Math+Physics | 0.040 | 0.240 | -0.002 | 0.061 |
| Medicine+Physics | -0.018 | -0.053 | **0.254** | **0.078** |

