# OpenReview forum: "Complementarity: Toward Better Metrics and Optimizing Data Efficiency in LLMs"
_TMLR — Accepted by TMLR_

### Review · Reviewer_QQi6 · 2025-06-23

**Summary Of Contributions:**

This paper proposes Complementarity, a metric that correlates with domain-specific task performance and measures the impact of fine-tuning on one domain's data on performance in a target domain. It is designed to be efficient and generation-free, and the authors show that it can guide data selection strategies that reduce training data while maintaining or improving performance across a number of  domains.

**Audience:**

Yes

**Claims And Evidence:**

No

**Requested Changes:**

- “We also demonstrate that models trained on data selected by Complementarity outperformed those trained on data selected by traditional evaluation metrics, exhibiting the strength of the metric.” The most common way to evaluate and select LLMs are perplexity scores, I would suggest to compare against standard measures of perplexity compared to Complementarity.

- The paper lacks a sound theoretical explanation for why changes in perplexity (via Complementarity) are expected to correlate with downstream task performance, especially for tasks requiring reasoning beyond surface fluency/language generation abilities.

- To me it appears that a lot of the discussion on accuracy being a limited metric are flawed comparisons, as standard
 classifiers are trained on cross-entropy losses and accuracy rather used for human-understandable evaluation.

- Include justification of P@1 ratios as the main comparison is missing. Please clarify why this metric is considered over alternatives like standard accuracy, F1, recall, etc.

- Introduce the mostly lacking discussion of limitations, e.g. potential limitations or failure cases, for example, domains where Complementarity may not predict performance well, or where interference can be expected.

- Introduce synthetic data settings, e.g. clearly distinct domains or controlled data mixtures that could help isolate and validate the specific behavior of Complementarity in a clean, interpretable way.

- Provide experimental details, refer to datset tables in the Appendix.

**Strengths And Weaknesses:**

**Strengths**
- The paper is overall well-written, easy to follow and addresses a relevant challenge.
- The empirical results appear reproducible and technically sound (with some potential flaws in experimental design).
- The metric is used to guide the selection of training subsets that can outperform full-dataset training, which may reduce data and compute requirements.


**Weaknesses**
- “When taking the entire table into account [(Table 1)], there is an average correlation of 0.7603 [...], which represents an incredibly strong p-value of <0.00001. This constitutes support for Complementarity being an excellent predictor of performance” This is a very strong statement given the limited support and theoretical connection of Complementarity and other metrics. Complementarity measures the log ratio of perplexities on some tuned model compared to the base model, indicating that increasing general language modeling abilities of the tuned model should correlate with increased down-stream task performance.
- Table 2: “The second section presents the Complementarity on the same test sets for models fine-tuned on training domain pairs.” Did the authors control for data quantity?
- Related works are mostly quite recent works, yet this issue of domain transfer and generalization is one of the key areas of ML research with a long-standing history of published works and concepts. More connections to existing works are missing here.

Overall, I think some of the ideas presented in this paper are interesting and relevant, but the proposal of Complementarity lacks deeper theoretical motivations and novelty. The central claims are thus not sufficiently well supported, e.g. *why* does Complementarity work better than pure perplexity or accuracy? I think the presented evidence supports some correlation to downstream task performance though

---

> ### Author Response · Authors · 2025-07-01
> **Response to Reviewer**
>
> # Author Response
>
> Dear reviewer, thank you for your review!
>
> ---
>
> ## Requested Changes
>
> > **Reviewer comment**
> >  The most common way to evaluate and select LLMs are perplexity scores, I would suggest to compare against standard measures of perplexity compared to Complementarity.
>
> **Response**
>
> That's exactly the point! The goal of this paper is to test the capabilities on specific tasks, not in general language fluency. As such, we made sure to use four key benchmarks from among the most common and popular across the field, as is presented in Table 1. This is newly discussed in the text in red, please see the last few red lines of page 5 and the remainder of the same red text continuing to page 6.
>
> > **Reviewer comment**
> > The paper lacks a sound theoretical explanation for why changes in perplexity (via Complementarity) are expected to correlate with downstream task performance, especially for tasks requiring reasoning beyond surface fluency/language generation abilities.
>
> **Response**
>
> Thank you for the great advice. We added a paragraph in red to explain this with a few relevant references. Please see the first red paragraph on page 5.
>
> > **Reviewer comment**
> >  To me it appears that a lot of the discussion on accuracy being a limited metric are flawed comparisons, as standard classifiers are trained on cross-entropy losses and accuracy rather used for human-understandable evaluation.
>
> **Response**
>
> As LLMs are generative models, rather than classifiers, the usage of classification-oriented metrics (such as accuracy) is difficult, given that it is built around the notion of correctness. To extract 'correctness' from an LLM requires a well-defined task and a lengthy pipeline for answer extraction. For more details, please see four paragraphs of page 3, the first starting with the words "The second issue".
>
> > **Reviewer comment**
> >  Include justification of P@1 ratios as the main comparison is missing. Please clarify why this metric is considered over alternatives like standard accuracy, F1, recall, etc.
>
> **Response**
>
> Measures such as accuracy, F1, recall, etc. rely on classification, namely on the ability to classify correctly vs. incorrectly and thus have four different option: true or false positives, and true or false negatives. For this reason, these are ill-suited to use in generative models like LLMs.
>
> > **Reviewer comment**
> >  Introduce the mostly lacking discussion of limitations, e.g. potential limitations or failure cases, for example, domains where Complementarity may not predict performance well, or where interference can be expected.
>
> **Response**
>
> You're right, thank you for pointing this out! This is now discussed in the text, see the second red paragraph on page 5 and the red text on page 12.
>
>
> > **Reviewer comment**
> >  Introduce synthetic data settings, e.g. clearly distinct domains or controlled data mixtures that could help isolate and validate the specific behavior of Complementarity in a clean, interpretable way.
>
> **Response**
>
> To have a significant impact on models such as Mistral (which has more than 7 billion parameters), such a database would need to be quite large. It is difficult to create such large synthetic databases which are clearly distinct. As such, we choose to focus on relevant real-world databases with practical applications, and made sure to clearly explain the details of the databases selected and their representative properties, as referenced in the response to the first requested change.
>
>
> > **Reviewer comment**
> >  Provide experimental details, refer to datset tables in the Appendix.
>
> **Response**
>
> Thank you for noticing! The dataset tables are referenced in the text, and the experimental section was reworked, see the middle of page 9.
>
> > **Reviewer comment**
> > Related works are mostly quite recent works, yet this issue of domain transfer and generalization is one of the key areas of ML research with a long-standing history of published works and concepts. More connections to existing works are missing here.
>
> **Response**
>
> The fact that many are recent works reinforces the fact that it is still an important area of research. On the other hand, if you have any specific recommendations for references, we would appreciate and be happy to include them.
>
> ---
>
> Once again, thank you so much for the insightful and helpful comments!

---

### Review · Reviewer_cjrA · 2025-06-24

**Summary Of Contributions:**

This paper introduces a new metric called Complementarity, designed to quantify how fine-tuning a language model on data from one domain affects performance on tasks from another domain. The metric is based on the change in perplexity of a target domain dataset before and after fine-tuning on a source domain. They further propose a data selection algorithm based on Complementarity. The paper makes the case that Complementarity is a low-cost method of seeing how fine-tuning on some data may impact performance on data from other domains before actually doing any fine-tuning.

**Audience:**

Yes

**Claims And Evidence:**

Yes

**Requested Changes:**

* Provide at least some intuitive or theoretical discussion of why Complementarity appears pseudo-symmetric. Even a hypothesis or link to prior work on representation similarity or domain alignment would strengthen the result.
* Discuss generalizability beyond problem-solving tasks. What limitations might arise in applying Complementarity to open-ended tasks like summarization, dialogue, or creative generation? How about cross-lingual applications?
* Add a brief analysis of the domain pairs (e.g., why do Physics and Medicine help each other?), or at least clarify whether the observed results are expected or surprising. Currently, the paper reports findings without much deeper interpretation.

**Strengths And Weaknesses:**

Strengths
* The central idea is creative, well-motivated, and novel. While perplexity-based comparisons are known, formalizing them into a reusable metric for inter-domain data utility is, to my knowledge, new.
* The experiments are thoughtfully designed and executed with clarity: multiple domains, ablation across domain pairs, and validation-driven data selection.
* Complementarity has clear practical appeal, since it avoids generation, should be task-agnostic, and offers a scalable alternative to manually tuned metrics or domain heuristics.

Weaknesses
* Only 4 domains are explored, and all of them involve structured problem-solving tasks. There is no evaluation on classic language understanding or generative tasks, so it's unclear whether Complementarity generalizes beyond this setting.
* The paper emphasizes the empirical correlation between Complementarity and performance, but offers little theoretical analysis or explanation for why the metric works.
* A notable empirical finding is the pseudo-symmetry of Complementarity between domains (e.g., A→B ≈ B→A), but this is treated as an observation only; no intuition or follow-up analysis is provided, which feels like a missed opportunity.
* Overall, the work feels more like a clever and promising insight than a fully matured method. The framing is elegant, but the scientific depth, both in analysis and scope of evaluation, is somewhat limited.

---

> ### Author Response · Authors · 2025-07-01
> **Response to Review**
>
> # Author Response
>
> Dear reviewer, thank you for your review!
>
> ---
>
> ## Requested Changes
>
> > **Reviewer comment**
> >  Provide at least some intuitive or theoretical discussion of why Complementarity appears pseudo-symmetric. Even a hypothesis or link to prior work on representation similarity or domain alignment would strengthen the result.
>
> **Response**
>
> This is an excellent point, and we added a paragraph discussing this in the text. It is in red in the text, but we copy it below for convenience:
>
> "Pseudo-symmetry is a useful analytic property, and indicates that Complementarity represents a fundamental attribute of the underlying data, rather than the specifics of the model. Linguistic domains, particularly in high-dimensional representations, are known to generally cluster into defined regions of the representation space \citep{aharoni-goldberg-2020-unsupervised, gururangan2023scalingexpertlanguagemodels}. Although Complementarity is not fully symmetric due to differences across domain pairs regarding utilization of tokens, prevalence of n-grams, etc., the process of learning to align language with the target structure and gain fluency in a particular section of the language space is a bi-directional process, such that training on either domain will similarly impact Complementarity."
>
> > **Reviewer comment**
> > Discuss generalizability beyond problem-solving tasks. What limitations might arise in applying Complementarity to open-ended tasks like summarization, dialogue, or creative generation? How about cross-lingual applications?
>
> **Response**
>
> We feel that the extension of Complementarity to open-ended tasks is an excellent avenue to explore in future work. We now list this and briefly explain it in the future work section, copied below for convenience:
>
> "Additionally, the burden of answer extraction design makes certain datasets and domains infeasible for evaluation, such as creative writing and poetry which have no singular correct response. Since Complementarity is not a binary metric and rather generally measures fluency, we hypothesize that this difficulty may be somewhat mitigated by the use of Complementarity."
>
>
> > **Reviewer comment**
> >  The paper emphasizes the empirical correlation between Complementarity and performance, but offers little theoretical analysis or explanation for why the metric works.
>
> **Response**
>
> Thank you for the great advice. We added a paragraph in red to explain this with a few relevant references, copied below:
>
> "However, there is good reason to think Complementarity may inform model success. First, perplexity on prompts has been recently demonstrated to correlate strongly with performance for both multiple-choice and open-ended tasks, and according to some research even more predictive of model behaviors than model size or training computation \citep{xia2023trainingtrajectorieslanguagemodels, gonen2024demystifyingpromptslanguagemodels}. Second, the structure of Complementarity closely resembles an LLM version of expected information gain \citep{EIG}, which has seen great success in decisions trees \citep{nowozin2012improvedinformationgainestimates} and is experiencing ongoing refinement in Bayesian modeling \citep{smith2023predictionorientedbayesianactivelearning}. Although expected information gain is a theoretical value that requires estimation, Complementarity is strictly empirical across the measured data, and thus is highly efficient even in the field of generative AI \citep{Goda_2019, li2025expectedinformationgainestimation}."
>
> > **Reviewer comment**
> >  Add a brief analysis of the domain pairs (e.g., why do Physics and Medicine help each other?), or at least clarify whether the observed results are expected or surprising. Currently, the paper reports findings without much deeper interpretation.
>
> **Response**
>
> This is an excellent point, thank you! We added a couple paragraphs to discuss this in the text. For lack of response characters, we will refer you to the text (text in red on Page 7, and second red paragraph on Page 8).
>
>
> > **Reviewer comment**
> >  Only 4 domains are explored.
>
> **Response**
>
> It is true that only four domains were explored, but they were carefully selected to be representative and have a number of important varied properties. We explain this in the text in the red text at the very bottom of page 5 extending to the end of said paragraph in page 6.
>
> ---
>
> Once again, thank you so much for the insightful and helpful comments! We really enjoyed and appreciated your comments, and feel that the paper is much improved having incorporated them!

---

> > ### Comment · Reviewer_cjrA · 2025-07-09
> > **Acknowledgment of Author Response**
> >
> > Thank you to the authors for their response. I believe the new changes strengthen the paper.

---

### Review · Reviewer_gHZq · 2025-06-26

**Summary Of Contributions:**

The authors propose a novel and computationally efficient metric named Complementarity that measures the effect of fine-tuning a model on datasets from training domains on a target domain. Some experiment shows that the proposed Complementarity metric has a strong correlation with P@1 metric. As it measure shift in performance, they also propose a data selection algorithm based on this measure. Experiments show that tailored dataset based on this algorithm improves model performance over baselines on multiple domains.

**Audience:**

Yes

**Broader Impact Concerns:**

I do not have any concerns for broader impact.

**Claims And Evidence:**

No

**Requested Changes:**

1. Writing needs improving including (but not limited to) the parts I mentioned.
2. There should be clearer explanation for weakness2
3. It would be better to tone down and remove some parts of Section 2 to avoid impression that Complementarity metric is demonstrated to be useful for all cases introduced in Section 2.
4. Some theoretical support would be helpful to understand why Complementarity is a good metric to check, or at least some intuition should be provided.
5. Analytic or empirical comparison in computational overhead would be helpful for readers to understand what is the cost for the performance improvement.
6. It would be more convincing if there are experiment results with different LLMs other than Mistral 7B Instruct v0.

Questions:
1. Doesn’t P@1 stand for precision at 1? How does it go higher than 1?

**Strengths And Weaknesses:**

Strengths
1. It provides good description for the cases where current metrics are limited.
2. It provides good demonstrations that show the effectiveness of Complementarity metrics although it is not comprehensive as stated in the weakness3 below.
3. The proposed method is simple but fast and effective

Weakness
1. Writing is a bit confusing. For example, it is hard to understand what $PP(D_j)$ (respectively $H(D_j)$) means, and P@1 ratio is not properly defined; for some readers who are not familiar with the field, it may be hard to understand.
2. The explanation for having the second values for correlation in Table 1 is not convincing. How and why conditional domain dependence between coding training dataset and math evaluation dataset matters?
3. I had impression that the paper is overselling the proposed metric. Section 2 introduces limitation of the current metrics in four different scenarios. Section 3 demonstrates that the proposed Complementarity metric has high correlation with P@1, which shows potential that it can be a good alternative for the case of (a) and (b). However, there is no demonstration for the case of (c) and (d). It is hard to infer that Complementarity metric would be a good surrogate for (c) and (d) based on demonstrations in Section 3.
4. There is no theoretical support or intuition behind why Complementarity shows how correlation with a target metric.
5. The computational overhead of the data selection scheme based on Complementarity seems enormous compared to the improvement reported.
6. As far as I understood, experiments are done only with Mistral 7B Instruct v0.

---

> ### Author Response · Authors · 2025-07-01
> **Response to Reviewer**
>
> # Author Response
>
> Dear reviewer, thank you for your review!
>
> ---
>
> ## Requested Changes
>
> > **Reviewer comment**
> >  Writing needs improving including (but not limited to) the parts I mentioned.
>
> **Response**
>
> We made a significant effort to clean up the language and be more precise with explanations, particularly focus on the areas you mentioned. In particular, removed the H(D), explained P@1 in depth, etc.
>
> > **Reviewer comment**
> >  There should be clearer explanation for weakness2
>
> **Response**
>
> We added a paragraph in red to explain this, copied below:
>
> "There is one key issue to note. As with all perplexity-based metrics, Complementarity is sensitive to domain dependence. For example, assume we have a model trained on a coding dataset, which includes some mathematics-like expressions, such as "n = 5", "x = x + 1", "if 1 == 2". Such examples naturally familiarize the model with such structure, and will therefore lower perplexity (i.e. raise complementarity) on a mathematics dataset, even that performance will likely be lowered, given the irrelevance of what was learned to mathematics. The reverse direction would not hold, as a mathematics database contains only a small percentage of example with structure relevant to coding. On the other hand, training on both mathematics and coding teaches the model to distinguish the finer features of both structures, and thus apply the correct knowledge to each problem, so Complementarity is particularly strong in the multi-domain case."
>
> > **Reviewer comment**
> >  It would be better to tone down and remove some parts of Section 2 to avoid impression that Complementarity metric is demonstrated to be useful for all cases introduced in Section 2.
>
> **Response**
>
> Agreed and done. Item (d) of section 2 was removed, and offered as a lane of future work in the discussion.
>
> > **Reviewer comment**
> >  Some theoretical support would be helpful to understand why Complementarity is a good metric to check, or at least some intuition should be provided.
>
> **Response**
>
> Thank you for the great advice. We added a paragraph in red to explain this with a few relevant references, copied below:
>
> "However, there is good reason to think Complementarity may inform model success. First, perplexity on prompts has been recently demonstrated to correlate strongly with performance for both multiple-choice and open-ended tasks, and according to some research even more predictive of model behaviors than model size or training computation \citep{xia2023trainingtrajectorieslanguagemodels, gonen2024demystifyingpromptslanguagemodels}. Second, the structure of Complementarity closely resembles an LLM version of expected information gain \citep{EIG}, which has seen great success in decisions trees \citep{nowozin2012improvedinformationgainestimates} and is experiencing ongoing refinement in Bayesian modeling \citep{smith2023predictionorientedbayesianactivelearning}. Although expected information gain is a theoretical value that requires estimation, Complementarity is strictly empirical across the measured data, and thus is highly efficient even in the field of generative AI \citep{Goda_2019, li2025expectedinformationgainestimation}."
>
> > **Reviewer comment**
> >  Analytic or empirical comparison in computational overhead would be helpful for readers to understand what is the cost for the performance improvement.
>
> **Response**
>
> We rewrote the algorithm section (to separate the algorithm from the evalution experiment). The amount of training required is roughly comparable to the full model, albeit more parallelizable. Broadly, given 100 splits of the dataset, we train models on each of the splits (in parallel, taking the same compute as the full model), run complementarity evaluation (incredibly fast and efficient), and then fine-tune once more on the best k splits, so overall would result in (100+k)/100 of the compute required for the full model.
>
> > **Reviewer comment**
> >  It would be more convincing if there are experiment results with different LLMs other than Mistral 7B Instruct v0.
>
> **Response**
>
> We definitely agree that this point needed some more explanation. We used Mistral as a representative "average" for LLMs, and added a section to explain and discuss this in red in the text, copied here:
>
> "For a model choice, we looked for a highly representative model, so that results will naturally generalize to many other common LLMs: one which is strong, but not over-tuned; large enough to be useful, but not so large it is difficult to deploy; one which is open-source and the base of many popular fine-tuned models; and finally is highly efficient under quantization, so it can be fine-tuned in a realistic time-scale. As such, we settled on Mistral 7B Instruct v0.2 \citep{jiang2023mistral7b}."
>
> ---
>
> Once again, thank you so much for the insightful and helpful comments!

---

### Decision · Action_Editor_4pES · 2025-08-12

**Recommendation:** Accept with minor revision

**Additional Comments:**

Based on reviewer feedback, I expect the authors to execute at least one of these additional experiments:

 - Add at least partial results from one other different size model. This would strengthen the generalizability of the claims, and is particularly important given that here we are talking about an evaluation metric, and the claims made.
 - Demonstrate the metric's utility on at least one open-ended task, as suggested by reviewers. Even if not done, the proposed paragraph is very vague. For example, I don't quite understand how non-binary metric and measure of fluency are opposing. Please revise this, making it more concrete.

In addition, the authors are expected to carefully review all the broad claims made (like the one mentioned above), and revise to make them more specific to the explored setup (e.g., types of tasks tested one). I will be looking for these changes when reviewing the final submission.

**Audience:**

Yes

**Audience Explanation:**

All the reviewers agreed that the proposed metric for evaluating LLMs would be of interest to TMLR audience, and could potentially serve as inspiration for future work in this direction.

**Claims And Evidence:**

Yes

**Claims Explanation:**

There has been some disagreement among the reviewers, but looking over the detailed comments, I would say that the experimental scope is largely sufficient to support most of the claims made in the submission. At the same time, there is space for improvement. For example, as noted by one of the reviewers, the claim that "Complementarity is an excellent metric" is quite strong, especially given somewhat limited evidence. Theoretical justification is also missing.

I therefore expert that the authors will change the language here, and instead of using such broad claims they can be more specific.

In general, I think the bar for adding a new evaluation metric should be quite high. The submission makes an interesting proposal, but the depth and breath are somewhat limited. At the same time, the reviewers thought that the proposed ideas are quite interesting, the existing experimental evidence is positive, and the work could inspire new research.